# Core–Shell Structured Phenolic Polymer@TiO_2_ Nanosphere with Enhanced Visible-Light Photocatalytic Efficiency

**DOI:** 10.3390/nano10030467

**Published:** 2020-03-05

**Authors:** Xiankui Xu, Lei Zhang, Shihua Zhang, Yanpeng Wang, Baoying Liu, Yanrong Ren

**Affiliations:** Henan Engineering Laboratory of Flame-retardant and Functional Materials, Institute of Functional Polymer Composites, College of Chemistry and Chemical Engineering, Henan University, Kaifeng 475004, China; 104753170806@henu.edu.cn (X.X.); zhangl@henu.edu.cn (L.Z.); 104753190744@henu.edu.cn (S.Z.); ypwang@henu.edu.cn (Y.W.); liubaoying666@163.com (B.L.)

**Keywords:** phenolic polymer, core–shell structure, titanium dioxide, photocatalyst

## Abstract

Core–shell structured TiO_2_ is a promising solution to promote the photocatalytic effectiveness in visible light. Compared to metal or semiconductor materials, polymers are rarely used as the core materials for fabricating core–shell TiO_2_ materials. A novel core–shell structured polymer@TiO_2_ was developed by using phenolic polymer (PP) colloid nanoparticles as the core material. The PP nanoparticles were synthesized by an enzyme-catalyzed polymerization in water. A subsequent sol–gel and hydrothermal reaction was utilized to cover the TiO_2_ shell on the surfaces of PP particles. The thickness of the TiO_2_ shell was controlled by the amount of TiO_2_ precursor. The covalent connection between PP and TiO_2_ was established after the hydrothermal reaction. The core–shell structure allowed the absorption spectra of PP@TiO_2_ to extend to the visible-light region. Under visible-light irradiation, the core–shell nanosphere displayed enhanced photocatalytic efficiency for rhodamine B degradation and good recycle stability. The interfacial C–O–Ti bonds and the π-conjugated structures in the PP@TiO_2_ nanosphere played a key role in the quick transfer of the excited electrons between PP and TiO_2_, which greatly improved the photocatalytic efficiency in visible light.

## 1. Introduction

Titanium dioxide (TiO_2_) is one of the most extensively investigated metal oxides due to its fascinating features, such as low cost, polymorphs, good chemical and thermal stability, and excellent electronic and optical properties [1,2,3]. These features render TiO_2_ highly promising in photocatalysts, dye-sensitized solar cells, energy storage, and biotechnology [4,5,6,7,8]. However, their performance is greatly limited by the wide energy band gap of approximately 3.2 eV, rapid electron–hole recombination, and relatively poor charge transport property [9,10,11]. To overcome these intrinsic drawbacks, various efforts were made to modify TiO_2_ materials. Doping with different elements, coupling with organic dyes, and composing with various materials were approaches extensively investigated to extend the active spectrum [12,13,14,15,16,17,18].

Recently, core–shell structured nanomaterials attracted considerable attention as they consist of different functional components integrated into one unit. These nanomaterials show improved physical and chemical properties, which are unavailable from the isolated components [19,20,21]. The active interfaces between individual components within a core–shell structure might give rise to outstanding synergistic functions and new properties. Thus, another promising solution for improving TiO_2_ photocatalyst efficiency under the visible spectrum was developed by assembling them into a core–shell structure. These structures are usually obtained by depositing TiO_2_ on a narrow-band-gap material to facilitate effective charge separation and improve the photostability. Many investigations focused on coupling TiO_2_ with metal or semiconductor core [22,23,24,25,26,27,28]. Despite the fact that polymers typically have low cost, easy preparation, and controllable size, research on core–shell structures of polymer and TiO_2_ remains quite limited. Polystyrene (PS) nanoparticles are frequently used as the core due to various synthesis methods, tunable size, and easy functionalization of surface. Imhof utilized colloidal PS spheres uniformed with poly(vinylpyrrolidone) (PVP) as the core and coated the spheres with a well-defined layer of amorphous TiO_2_ [29]. Liu et al. used acrylic acid as the comonomer to prepare a PS mini-emulsion, and they obtained core–shell PS and TiO_2_ microspheres via the strong interaction between TiO_2_ and carboxyl groups [30]. Wang and co-workers prepared a core–shell PS@TiO_2_ photocatalyst via vapor phase hydrolysis. They found that the core–shell catalyst had relatively better activity in the degradation of methylene blue (MB) than bare TiO_2_, and the PS core may have consumed some active radicals, which resulted in bad recycling [31]. Türk et al. prepared PS colloidal particles with TiO_2_ coating via a sol–gel process with in situ hydrolysis, and the core–shell particles revealed good photocatalytic activity for the oxidation of 4-methoxybenzyl alcohol with O_2_ in water [32]. Wu’s group prepared PS@TiO_2_ core–shell particles using a layer-by-layer self-assembly method. They revealed that the PS@TiO_2_ materials obtained via this method can be used to degrade rhodamine B (RB) in HCl solution, and the degradation rate of RB increased with the increase in TiO_2_ colloid shell layer [33].

Recently, the efficient enzymatic polymerization of phenol in aqueous solution was developed in the presence of templates, such as poly (ethylene oxide) (PEG) [34,35], cyclodextrin derivates [36,37], or surfactant [38,39]. The polymerization is fast and moderate without toxic formaldehyde. Generally, the structure of phenolic polymer (PP) prepared via enzymatic polymerization is composed of a mixture of phenylene and oxyphenylene. Thus, a great number of hydroxyl groups are presented on the surface of the polymer colloid particles, which grant high reactivity.

In this work, a PP@TiO_2_ core–shell nanosphere was prepared via the sol–gel method. PP was synthesized by a horseradish peroxidase-catalyzed phenol polymerization with PEG in water. The PP colloid particle was used as the core and support substrate of the photocatalyst. The thickness of the TiO_2_ shell was controlled by the amount of the precursor of TiO_2_. The chemical structure was investigated to understand the interaction between the core of PP and the shell of TiO_2._ The photocatalytic activity was determined by degradation of rhodamine B (RB) solution under visible- light irradiation.

## 2. Materials and Methods

### 2.1. Materials

Horseradish peroxidase (HRP) (RZ = 2.5, activity = 200 U/mg) was purchased from Shanghai Guoyuan Biotechnology Co., Ltd. (Shanghai, China) and used without further purification. Poly (ethylene oxide) (PEG) was obtained from Tianjin Guangfu Fine Chemical Research Institute (Tianjin, China). Tertrabutyl titanate (TBOT) was purchased from Tianjin Kemiou Chemical Reagent Co., Ltd. (Tianjin, China). Hydrogen peroxide (30%) was obtained from Luoyang Haohua Chemical Reagent Co., Ltd. (Luoyang, China). All other chemicals employed in this work were obtained from various commercial suppliers and were of the highest purity available.

### 2.2. Measurements

^1^H-NMR spectra were recorded on a Bruker DPX400 spectrometer (Bruker, Zurich, Switzerland). Fourier-transform infrared (FT-IR) spectra were obtained on an Avatar 360 spectroscope (Bruker, Ettlingen, Germany). Gel permeation chromatography (GPC) measurements were conducted with a water 410 GPC (Waters, Milford, PA, USA) equipped with Waters styragel column (HT4 + HT3) using THF as the eluent; the molecular weights were calibrated with polystyrene standards, and the flow rate was set at 1.0 mL/min at 35 °C. Dynamic light scattering (DLS) measurement was performed using Nanotrac Wave II (Microtrac, Montgomeryville, PA, USA), and the scattering angle was fixed at 180°. The surface morphology of samples was analyzed on a JSM-7610F (JEOL, Tokyo, Japan) scanning electron microscope (SEM), and the main elements were measured by energy-dispersive X-ray spectrometry (EDS) (JEOL, Tokyo, Japan). The samples on the silicon wafers were mounted rigidly to a copper specimen holder using a conductive adhesive. Transmission electron microscopy (TEM) studies were performed on a JEM-2100 electron microscope (JOEL, Tokyo, Japan) operating at an acceleration voltage of 100 kV. The nanoparticle solution was dropped on copper grids and dried at room temperature. X-ray diffraction (XRD) was recorded on a D8 Advance X-ray diffractometer (Bruker, Karlsruhe, Germany) with CuK_α_ radiation. X-ray photoelectron spectroscopy (XPS) was performed using a Thermo EscaLab 250Xi photoelectron spectrometer (Thermo Fisher Scientific, West Sussex, UK). Electron paramagnetic resonance (EPR) spectra were recorded on a Bruker A300 spectrometer (Bruker, Karlsruhe, Germany) at ambient temperature. UV–Vis absorption spectra were recorded using a U4100 spectrophotometer (Hitachi, Shanghai, China).


**Enzyme-catalyzed Polymerization of phenol**


A typical run was as follows: phenol (0.47 g, 5.0 mmol) and PEG (0.22 g, 5.0 mmol of monomer unit) were dissolved in 45 mL of water. Then, the enzyme solution of HRP (2.0 mg in 5 mL of water) was added. To this solution, 3.4 mL of 5% hydrogen peroxide aqueous solution was added dropwise for 1 h. The mixture was stirred at room temperature in air for 30 min. A brown emulsion of phenolic polymer (PP) was obtained. For characterization, the as-prepared polymer was collected by centrifugation and washed with water repeatedly, followed by drying in vacuum.


**Preparation of the PP@TiO_2_ core–shell nanosphere**


The PP@TiO_2_ core–shell nanospheres were prepared via a sol–gel and hydrothermal reaction. Typically, 25 μL of TBOT was firstly dissolved in 5 mL of ethanol (solution A). Then, a 1 mL emulsion of PP was mixed with 10 mL of ethanol (solution B). Then, solution B was added to solution A and kept at 80 °C for 1 h with vigorous stirring. After centrifugation and drying, a yellow powdery product was obtained. Then, 100 mg of powder was mixed with 20 mL of deionized water and transferred to a 100 mL Teflon autoclave. The mixture was heated to 180 °C, maintained for 8 h. The reactor was cooled down to room temperature naturally. The resulting nanosphere was collected by centrifugation and washed with ethanol and water repeatedly.


**Photocatalytic measurement**


The visible-light photocatalytic activity of the PP@TiO_2_ core–shell nanosphere was evaluated by the degradation rate of rhodamine B (RB) with an initial concentration of 20 mg/L. In a typical photodegradation experiment, 40 mL of RB solution and the sample photocatalyst containing 20 mg of TiO_2_ were placed in a 50 mL breaker. Before irradiation, the suspension was magnetically stirred in the dark for 2 h to reach the adsorption–desorption equilibrium between dye and photocatalyst. The light source was a 300 W Xe lamp equipped with an ultraviolet cut-off filter (λ > 400). The average visible-light intensity measured by the radiometer was 20 mW/cm^2^. In this experiment, a 20 mg/L aqueous RB solution was mixed with 0.6 g/L photocatalyst powder. RB concentration was determined using UV–Vis absorption at defined time intervals. For the purpose of comparison, P25, PP, and pure TiO_2_ samples were also used to degrade RB.

## 3. Results

### 3.1. Characterizaion of PP

The enzyme-catalyzed polymerization of phenol in water is a typical precipitation polymerization. During polymerization, the powdery polymer precipitates from solution [34]. Serving as the core material, PP is expected to exhibit stability in water and controlled size. Thus, the polymerization was directly performed in water without solvents or buffer solution. We found that a homogeneous emulsion was formed by controlling the molecular weight of PEG. Different molecular weights of PEG, from 400 g/mol to 4000 g/mol, were used in the enzyme-catalyzed polymerization. The as-prepared emulsions were kept for several weeks, and the emulsion prepared with PEG 2000 showed the best stability. Thus, PEG 2000 was selected for the preparation of PP@TiO_2_ core–shell nanosphere. The morphology and particle size distribution of PP were analyzed by SEM and DLS. The SEM image is shown in Figure 1a. It was clear that PP particles presented a uniform spherical structure with a smooth surface and a diameter of about 180 nm. The optical photograph of the PP emulsion was a little brown and remained stable for several weeks (inset of Figure 1a). The size distribution curve (Figure 1b) from DLS analysis demonstrates that the PP presented as uniformly dispersed particles with size ranging from 120 nm to 250 nm, and the average size was 174 nm.

To investigate the chemical structure of PP, the polymer was separated from the emulsion by centrifugation. In the FT-IR spectrum (Figure 2a), a broad peak centered at 3400 cm^−1^ was ascribed to the vibration of the phenolic O–H bond. The peaks at 1597, 1488, 833, 754, and 692 cm^−1^ were characteristic of the various vibration modes of the C–H and C–C bonds of aromatic nuclei/rings. The peak at 1096 cm^−1^ corresponded to the symmetric vibration of the ether bond. The strong peak at 1209 cm^−1^ was due to the asymmetric stretching vibration of C–O–C and/or C–OH. The ^1^H-NMR spectrum (Figure 2b) of PP was measured in DMSO-d6. The single peak at 3.3 ppm was attributed to CH_2_CH_2_O, which means that some PEG still remained in the sample after washing with water. The broad peak at 6.6–7.4 ppm was attributed to aromatic units. The broad resonance signal at 9.2–9.6 ppm was the signal of hydroxyl groups. These results indicated that the structure of the polymer was composed of a mixture of phenylene and oxyphenylene (Ph/Ox) units. The ratio of Ph/Ox was determined by titration of the hydroxyl groups in the polymer, and it was found to be 24/76. The data of GPC showed that the number-average molecular weight was 1600 and the polymer dispersity index was 7.27.

### 3.2. Characterization of PP@TiO_2_ Core–Shell Nanosphere

The PP@TiO_2_ core–shell nanosphere was fabricated via the process shown in Scheme 1. After the preparation of a PP emulsion, the sol–gel process was utilized to form the PP@TiO_2_ core–shell nanosphere precursor. Then, the traditional hydrothermal treatment was selected to obtain crystalline TiO_2_ and prevent the core of PP from being damaged at high temperature. The reactive system simply included the core of PP and the shell of TiO_2_ without any additive or catalyst.

To investigate the effect of the shell thickness on the photocatalyst activity, PP@TiO_2_ core–shell nanospheres were prepared with different amounts of TBOT, i.e., 25 μL, 50 μL, 75 μL, and 100 μL. Figure 3 presents SEM images of the four nanospheres. After the hydrothermal reaction, a rough layer aggregated by the dense and uniform nanoparticles was formed on the surface of PP nanoparticles. The size of the nanoparticles increased as increasing the amount of TiO_2_ precursor. With 25 μL of TBOT, the size of the nanospheres was about 220 nm. When more TBOT was introduced into the nanosphere, the size rapidly increased. A nanosphere with a diameter of 340 nm was observed corresponding to 50 μL of TBOT. The size of the nanospheres reached up to 650 nm diameter when 100 μL of TBOT was used. The mapping pattern of PP@TiO_2_-50 is shown in Figure 3e. The element of C corresponded to PP, and the elements of O and Ti were due to TiO_2_. Therefore, it can be deduced that the nanospheres were successfully constructed from TiO_2_ and PP.

TEM was further used to observe the structure of the nanospheres, as shown in Figure 4. In the sample of PP@TiO_2_-25, a strong contrast between the dark edges and gray centers indicated the core–shell structure of the nanosphere. The PP nanoparticles were covered by an aggregation of TiO_2_ particles, and the thickness of the shell was about 30 nm. With the increase in TBOT amount, the thickness of the shell apparently increased and the size of the core of PP greatly reduced. The nanosphere prepared with 50 μL of TBOT showed a shell thickness of about 80 nm and a core diameter of about 150 nm. The shell thickness of the nanosphere prepared with 75 μL of TBOT was about 200 nm, and the core diameter was below 120 nm. With 100 μL of TBOT, the size of the core reduced to 50 nm with a shell of 280 nm. The results indicated that TBOT permeated into the core region when the amount of TBOT was high.

The X-ray diffraction (XRD) total pattern in Figure 5 was used to study the microstructure of the PP@TiO_2_ nanospheres. The typical X-ray diffraction spectrum of amorphous polymer is shown at 2*θ* = 20°. The XRD patterns of the PP@TiO_2_ nanospheres showed apparent diffraction peaks at the 2*θ* values of 24.9°, 37.5°, 47.5°, 54.1°, and 62.4° that were indexed to the (101), (004), (200), (105), (211), and (204) planes, matching well with the anatase TiO_2_ JCPDS card (no. 21-1272) [40]. A pure TiO_2_ sample was prepared via the same sol–gel and hydrothermal process as PP@TiO_2_. Comparing the diffraction peak of PP@TiO_2_-25 to pure TiO_2_, it was obvious that the peaks became narrow and the intensities increased. This indicates that PP can improve the crystallinity of TiO_2_, which is very beneficial for an improvement in the catalytic activity. However, the effect was gradually weakened when the TBOT amount increased. As the TBOT amount was increased to 100 μL, the pattern resembled that of pure TiO_2_.

More detailed information regarding the chemical structure of the PP@TiO_2_ nanospheres was obtained through characterization by FT-IR, XPS, and EPR. Figure 6a shows the FT-IR spectra of the pure TiO_2_ and PP@TiO_2_ samples before and after the hydrothermal reaction. For pure TiO_2_, the peaks at 3422 cm^−1^ and 1630 cm^−1^ corresponded to the stretching and bending vibration of OH bonds, which resulted from the physically absorbed water and surface hydroxyl groups. Compared with pure TiO_2_, the PP@TiO_2_ particle before hydrothermal reaction provided the characteristic peaks of PP and TiO_2_, which indicated that the nanospheres were composed of PP and TiO_2_. After the hydrothermal reaction, the peak at 1632 cm^−1^ strengthened due to the intramolecular hydrogen bonding between PP and TiO_2_, and a new band at 1210 cm^−1^ indicated the formation of new bonds. Some research reported the formation of covalent C–O–Ti bonds between polymers with hydroxyls and TiO_2_ [41,42]. Thus, the stronger peak suggested that PP had a covalent contact with TiO_2_. The result was confirmed by XPS. The high-resolution C *1s* XPS spectrum and the fitting curves are shown in Figure 6b. The major peak with a binding energy of 284.5 eV was attributed to the C–C and C–H bonds of PP. The peak at 285.6 eV was ascribed to the C–O–Ti bond. The broad peak centered at 288.4 eV was attributed to the –C=O bond, while the peaks around 292.3 eV belonged to π–π* bonds. The Ti *2p* peaks at 485.4 eV and 464.1 eV in Figure 6c were attributed to the Ti *2p_1/2_* and Ti *2p_3/2_* spin orbit splitting, while the peaks at 460.0 eV were ascribed to the C–O–Ti bond [9]. The O *1s* spectrum in Figure 6d displays two peaks at 529.7 eV and 532.2 eV, which corresponded to the Ti–O–Ti and C–O–Ti bonds, respectively [43]. Figure 6e demonstrates two bonds located at the binding energies of the EPR spectrum of the PP@TiO_2_ core–shell nanosphere, which was measured to confirm the unpaired electrons in this core–shell structure which play an important role in photocatalysis. As shown in Figure 6c, the lines for the pure TiO_2_ and PP presented a negligible signal peak, while PP@TiO_2_ showed a very strong EPR signal centered on the magnetic field strength of 580 G. The result indicated the formation of π-conjugated structures on the interface of the TiO_2_ shell, which were attributed to the delocalized π–π* electrons formed in the PP core (the inset in Figure 6c).

### 3.3. Photocatalytic Efficiency

The optical absorption of PP@TiO_2_ nanospheres was characterized by UV–visible spectra, as shown in Figure 7a. PP had a broad absorbance from 400 nm to 800 nm. Meanwhile, pure TiO_2_ showed a narrow absorption edge located at 389 nm with a band gap of 3.19 eV, which is consistent with the intrinsic bandgap absorption of anatase TiO_2_. When PP was composed with TiO_2_, it was obvious that all the samples extended their absorbance edges to the visible-light region, and the TiO_2_ amount significantly affected the optical property of visible-light absorption. PP@TiO_2_-25 had the strongest and longest absorbance edge of 520 nm, corresponding to a band gap of 2.38 eV. The curve of PP@TiO_2_-25 was similar to that of PP from 400 nm to 800 nm. This indicated that the shell of PP@TiO_2_-25 had no apparent hindrance of the visible-light absorption of PP. However, the absorbance of visible light of PP@TiO_2_ nanospheres gradually decreased with increased amount of TiO_2_. The absorbance edge of PP@TiO_2_-100 reduced to 400 nm, and its band gap rebounded to 3.1 eV. This result demonstrated that the TiO_2_ shell blocked the visible-light absorption of the nanosphere when the thickness exceeded some threshold value. The obvious decrease in the bandgap of PP@TiO_2_ may be attributed to the chemical bonding between TiO_2_ and PP with the formation of C–O–Ti bonds. The narrow band gap and long absorbance edge were beneficial for improving the visible-light photocatalytic efficiency.

To evaluate the photocatalytic efficiency of the as-prepared core–shell nanospheres, the photocatalytic degradation of RB aqueous solution under visible-light irradiation (λ > 400 nm) was investigated. All samples were suspended in the treated RB solution and stirred for 120 min without light to achieve the adsorption–desorption equilibrium. The normalized concentration (C/C_0_) of RB is shown in Figure 7b. C_0_ denotes the initial RB concentration of 20 mg/L. As PP was used as the photocatalyst, the concentration of RB was unchanged, suggesting that PP had no visible-light photocatalytic activity. Pure TiO_2_ had the similar effect to the commercial TiO_2_ material P25. Their degradation rates reached 50% after 4 h of visible-light irradiation. Compared to pure TiO_2_ and P25, all samples of PP@TiO_2_ nanospheres showed apparent adsorption of RB in the dark. The concentration of RB with PP@TiO_2_-25 decreased the most in comparison with the others. According to those results of SEM and TEM analysis, PP@TiO_2_-25 had the toughest surface and largest PP core, which endowed it with high adsorption capacity. After stirring for 120 min in the dark, there was 36%, 71%, 75%, and 82% RB remaining in the PP@TiO_2_-25, PP@TiO_2_-50, PP@TiO_2_-75, and PP@TiO_2_-100 samples, respectively. In spite of the absorption occurring in the first 30 min, the dark reaction lasted 120 min in order to eliminate the effect of absorption on the photodegradation. Under visible- light irradiation, all PP@TiO_2_ nanospheres were photocatalytically active. The removal yield presented great dependence on the amount of TiO_2_. PP@TiO_2_-25 obtained the best yield of 95% after 4 h of irradiation. Upon increasing the amount of TiO_2_, the removal yield gradually reduced. The removal yield of PP@TiO_2_-100 decreased to 63%.

The removal of RB clearly included adsorption and photodegradation. Removing the effect of adsorption during the darkness aspect, the Langmuir–Hinshelwood model was used to investigate the photocatalytic efficiency of PP@TiO_2_ nanospheres. The RB removal on these materials followed the pseudo-first order kinetic model shown in Figure 7c. The photocatalytic reaction can be interpreted by ln(*C/C_0_*) *= −kt* where *k* is the apparent rate constant with respect to the irradiation time *t*, and *C* and *C_0_* are the concentrations of RB at *t* and *t = 0* (time of irradiation), respectively [44,45]. The photocatalytic rate constant *k* of PP, P25, pure TiO_2_, and the samples PP@TiO_2_-25, -50, -75, and -100 were 0.12, 2.26, 2.46, 3.35, 3.10, 2.63, and 2.62 × 10^−3^ min^−1^, respectively. It was found that all core–shell nanospheres had a higher rate constant *k* than pure TiO_2_, especially PP@TiO_2_-25, with a thin TiO_2_ shell of 30 nm, which obtained the highest photocatalytic efficiency. This result can be attributed to the formation of the effective interface between the PP core and TiO_2_ shell in favor of efficient charge carrier separation.

To estimate the stability of the as-prepared photocatalyst, a five-cycle recycling experiment of PP@TiO_2_-25 was performed under visible-light irradiation, as shown in Figure 7d. The removal efficiency of RB on PP@TiO_2_-25 was 95.5% in the first cycle, followed by 96.7%, 95.4%, 94.5%, and 97.8% for the last four cycles. Thus, the photocatalyst did not exhibit any loss in removal activity in five cycles. This indicated that the PP@TiO_2_ nanosphere exhibited good cycle stability under visible- light irradiation.

On the basis of the above discussion, the fabrication of core–shell structured PP@TiO_2_ nanospheres enhanced the photocatalytic efficiency of TiO_2_ under visible-light irradiation. The mechanism of the electron transfer process between PP and TiO_2_ in the core–shell nanosphere is illustrated in Scheme 2. As the PP polymer is composed of phenylene and oxyphenylene units, there are a great number of hydroxyl and aromatic nuclei/rings on the surface of PP particles. During the hydrothermal process, the hydroxyl groups on the surface of the PP react with the hydroxyl groups of TiO_2_ to form C–O–Ti bonds. For TiO_2_, under visible-light irradiation, most of the photo-excited electrons and holes tend to rapidly recombine, and only a small number of them participate in the photocatalytic reaction, which results in a relatively low photocatalytic activity. On the other hand, for the core–shell-structured PP@TiO_2_ nanospheres with a thin TiO_2_ shell, the surface of the PP nanoparticles is covered by TiO_2_, which leads to the formation of a Z-scheme photocatalytic system between TiO_2_ and PP [46,47]. Under visible-light irradiation, the photo-generated holes tend to keep on the valence band (VB) of TiO_2_, while the electrons transfer to the VB of PP from the conduction band (CB) of TiO_2_ through the interfacial pathway of C–O–Ti bonds. The electrons in the VB of PP are further excited to its CB. This results in an efficient space separation of the photo-induced charge carriers. Then, the electrons stored in the CB of PP are trapped by O_2_ near the surface of PP, forming reactive superoxide radical ions O_2_^−^, while the holes in the VB of TiO_2_ react with adsorbed water molecules near the surface of TiO_2_, forming hydroxyl radicals ·OH. The subsequent oxidative and reductive reactions lead to the degradation of RB.

## 4. Conclusions

A novel polymer@TiO_2_ nanosphere, PP@TiO_2_, was prepared via the sol–gel method. Firstly, a stable PP emulsion was prepared through the enzyme-catalyzed polymerization of phenol in water in the presence of PEG. After the sol–gel and hydrothermal process, the core of PP was connected with the shell of TiO_2_ via C–O–Ti covalent bonds. The removal of RB under visible-light radiation showed that the photocatalytic efficiency of PP@TiO_2_ nanospheres is much better than that of pure TiO_2_ and commercial P25. The removal rate of RB increased upon decreasing the amount of TiO_2_. For PP@TiO_2_-25, with the lowest amount of TiO_2_, the removal rate of RB reached 95% within 240 min. Furthermore, the novel core–shell TiO_2_ nanosphere was shown to have good recycling properties.

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
