# Peer review of "Core–Shell Structured Phenolic Polymer@TiO2 Nanosphere with Enhanced Visible-Light Photocatalytic Efficiency"

_nanomaterials, 2020, doi:10.3390/nano10030467_

Round 1

Reviewer 1 Report

The paper presents an investigation on the preparation and testing of a semiconductor

core-shell structured TiO2, for the degradation of RB under visible light.

In my opinion, the paper needs of heavy revision before to be considered worth of publication in the requested journal

Of note, while a great part of the paper is devoted to the description of the synthesis, less attention is given to testing this material for the proposed application to degradation of RB, and to describe the reaction mechanism.

In particular:

The authors should give details about the reaction: Fig. 7 c is very poor, and the evaluation of the yield of different cycles is impossible to be read from it. Moreover, it should be explained the reason for the higher decrease in the concentration during the fifth cycle. Is there any modification of the structure during cycling? If yes, details should be given

Since adsorption of the reactant is observed during the process, a LH mechanism could be considered to interpret the kinetics

Regarding the proposed reaction mechanism, I do not understand how the PP could be active to the light, being it in the core of the particle. Moreover, as also assessed by the author themselves, it has a quite poor absorption capacity in the visible range. So, no proof of the proposed mechanism is given. Authors should also provide a more detailed mechanism, on the bases of the energetic levels of the different materials involved in the charge transfer process.

Finally, I suggest the authors to use the term “RB removal” rather than “RB degradation”. In fact, the removal of the reactant from the solution is not an indication of its degradation, because possible intermediates could be originated in the solution, but, in this work, they have not been identified.

Author Response

Response to Reviewer 1 Comments

Point 1: The authors should give details about the reaction: Fig. 7 c is very poor, and the evaluation of the yield of different cycles is impossible to be read from it. Moreover, it should be explained the reason for the higher decrease in the concentration during the fifth cycle. Is there any modification of the structure during cycling? If yes, details should be given. 

Response 1: Thanks for the helpful advice. The yield of different cycles is evaluated by testing the intensity of UV-Vis spectrum of the solution after configuration, without further operation. We have added the data of the different cycles. From the first to the last cycle, the removal rates are 95.5%, 96.7%, 95.4%, 94.5% and 97.8%, respectively. The yield in the fifth cycle is a little higher, which may be caused by the error in the recorded UV-visible spectra of RB during the removal reaction.

Point 2: Since adsorption of the reactant is observed during the process, a LH mechanism could be considered to interpret the kinetics.

Response 2: Thanks for the helpful advice. We noticed the obvious adsorption during the process without irradiation. The corresponding data of adsorption have been added in this section. The structure of PP core plays an important role on the absorption. We are working on the research of absorption property. Since this paper focused on the effect of the PP core on the photocatalytic activity of TiO2 shell, we made more discuss on the irradiation process.

Point 3: Regarding the proposed reaction mechanism, I do not understand how the PP could be active to the light, being it in the core of the particle. Moreover, as also assessed by the author themselves, it has a quite poor absorption capacity in the visible range. So, no proof of the proposed mechanism is given. Authors should also provide a more detailed mechanism, on the bases of the energetic levels of the different materials involved in the charge transfer process.

Response 3: Thanks for the helpful advice. We have done more discussion on the UV-visible spectra of this nanosphere. Compared with the spectrum of PP, the nanosphere with a thin TiO2 shell presents similar absorbance as the PP core from 400 nm to 800 nm. As the increase of the thickness of the shell, the absorbance gradually decreases. Thus, the shell thickness has a great effect on the photocatalytic efficiency. The corresponding discussion has been added in the “Result and Discuss Section 3.3”.

Point 4: I suggest the authors to use the term “RB removal” rather than “RB degradation”. In fact, the removal of the reactant from the solution is not an indication of its degradation, because possible intermediates could be originated in the solution, but, in this work, they have not been identified.

Response 4: Thanks for the helpful advice. The term “RB degradation” has been placed by “RB removal” in this paper.

Reviewer 2 Report

Authors reported pp@TiO2 nanosphere and studied their catalytic activity.  This work is interesting. However authors need to compare their outcomes with others works and improve the quality of manuscript to reach the standards of nanomaterials journal. Therefore, authors need to revise the manuscript.

  1. Authors need to index the XRD of TiO2 and PP phases.
  2. Add the XRD pattern for PP materials and compare with their nanocomposite.
  3. What is role of HRP? Why authors chosen this? Need to explain influence of HRP on their core-shell structure.
  4. Is this nanohybrid is stable enough? Authors need to synthesis without of HRP and compare their results with current derived core-shell structure.
  5. Add the XPS elements for Ti and O element and discuss in revised manuscript.
  6. Authors need explain the influence of NIR ? on catalytic activity.
  7. If possible studied the photocurrent and EIS analysis of nanocomposite.
  8. Scheme 2 is lack of band alignment of catalysts. Redraw with respect to band positions..
  9. Authors need to compare their outcome catalytic activities with literatures and please add the relevant article in revised manuscript: (i) Ceramics International 45 (2), 2178-2184; (ii) Ceramics International 44 (10), 11783-11791; (iii) Materials Research Bulletin 98, 314-321; and (iv) Journal of Alloys and Compounds 740, 574-586.
  10. Please rectify typo errors in manuscript.
  11. What is intensity of light used?

Author Response

Response to Reviewer 2 Comments

Point 1: Authors need to index the XRD of TiO2 and PP phases.

Response 1: The TiO2 JCPDS card has been added in Figure 5. PP is an amorphous polymer composed of phenylene and oxyphenylene units. The XRD spectrum of PP have been measured and added in Figure 5.

Point 2: Add the XRD pattern for PP materials and compare with their nanocomposites.

Response 2: The XRD pattern for PP had been added in Figure 5. The difference between PP and their nanocomposites has been discussed in the corresponding section.

Point 3: What is role of HRP? Why authors chosen this? Need to explain influence of HRP on their core-shell structure.

Response 3: HRP is abbreviation to horseradish peroxidase as we described in section 2.1. The peroxidase is used for enzyme-catalyzed polymerization of phenol monomer in water at room temperature. And the amount of HRP for polymerization is very small. In this method, the PP emulsion with uniform particle size can be obtained. The uniform PP particles are the template of the core-shell structure.

Point 4: Is this nanohybrid is stable enough? Authors need to synthesis without of HRP and compare their results with current derived core-shell structure.

Response 4: Yes, the nanohybrid is stable. The recycling-use testing has been proved the stability of this nanohybrid. HRP is enzyme catalyst for polymerization of PP in water. PP can not be polymerized in water without the enzyme.

Point 5: Add the XPS elements for Ti and O element and discuss in revised manuscript.

Response 5: Thanks for the helpful advice. The XPS spectra for Ti and O elements have been added in Figure 6. And the corresponding discuss have been added in Section 3.2.

Point 6: Authors need explain the influence of NIR ? on catalytic activity.

Response 6: The explaining has been added in Section 3.3 accordingly.

Point 7: If possible studied the photocurrent and EIS analysis of nanocomposite.

Response 7: Thanks for the helpful advice. However, the analysis is not available at this time.

Point 8: Scheme 2 is lack of band alignment of catalysts. Redraw with respect to band positions.

Response 8: The scheme 2 has been redrawn accordingly.

Point 9: Authors need to compare their outcome catalytic activities with literatures and please add the relevant article in revised manuscript: (i) Ceramics International 45 (2), 2178-2184; (ii) Ceramics International 44 (10), 11783-11791; (iii) Materials Research Bulletin 98, 314-321; and (iv) Journal of Alloys and Compounds 740, 574-586.

Response 9: These articles have been added in revised manuscript accordingly.

Point 10: Please rectify typo errors in manuscript.

Response 10: The rectify typo errors in manuscript has been revised carefully.

Point 11: What is intensity of light used?.

Response 11: The explanation of intensity of light had been written in Photocatalytic Measurement of Section 2.2.

Round 2

Reviewer 1 Report

The authors made some of the requested changes. However, some points of the discussion remain not clear. So that major revision is still needed.

In particular, the kinetics has to be better discussed: fig7c shows a clear effect of the adsorption which is still not included in the kinetic equation. Of note, figure is wrong, becouse the y axis cannot be positive if C/C0 decreases; neither it cannot start from 1! The Langmuir Hinshelvood mechanism shoud be claimed to justify the initial ternd of data, and the related combined equation used to model the data. Or, in alternative a different discussion has to be made regarding this aspect. (the following paper could be considered, for example: Separation and purification technol. (2109) 208, pp. 153-159)

Moreover, also regarding the description of the electronic mechanism, some doubts remain: if the energetic levels of the bands are correct, I do not understand how the electrons can migrate from the valence band of TiO2 to the PP. Maybe a different charge transfer mechanism has to be cited such as the Z mechanism. Neither the reactions occurring at the interface, leading to the final products are explained. I may suggest the authors to compare their model with literature (for example: RSC Advances (2016) 6(103), pp. 101671-101682)

For this reasons I cannot say that the paper, at this stage, is worth to be pubblished.

Author Response

Point 1: The kinetics has to be better discussed: fig7c shows a clear effect of the adsorption which is still not included in the kinetic equation. Of note, figure is wrong, because they axis cannot be positive if C/C0 decreases; neither it cannot start from 1! The Langmuir Hinshelwood mechanism should be claimed to justify the initial term of data, and the related combined equation used to model the data. Or, in alternative a different discussion has to be made regarding this aspect. (The following paper could be considered, for example: Separation and purification technol. (2019) 208, pp. 153-159)

Response 1: We great appreciate that the reviewer pointed out our wrong which should not be made. Fig7c has been redrawn according to the reference which was listed in the section of “References”. Because the adsorption reaction was deliberately prolonged, the process was not acceptable to include in the kinetic equation.  So we discussed the L-H kinetic equation with the concentration C0 at t0 the beginning of irradiation and the related equation was use to model the data. The explanation has been added in the corresponding section.

Point 2: Regarding the description of the electronic mechanism, some doubts remain: if the energetic levels of the bands are correct, I do not understand how the electrons can migrate from the valence band of TiO2 to the PP. Maybe a different charge transfer mechanism has to be cited such as the Z mechanism. Neither the reactions occurring at the interface, leading to the final products are explained. I may suggest the authors to compare their model with literature (for example: RSC Advances (2016) 6(103), pp. 101671-101682).

Response 2: Thanks for the helpful advice. After reading some literatures involved Z-mechanism including the one recommended by the reviewer, we think the Z-mechanism is right for the photocatalytic activity of PP@TiO2. So, the scheme 2 was redrawn, and the corresponding discussion was rewritten, and the corresponding literatures were listed in the section of “References”.

Reviewer 2 Report

Authors revised manuscript thoroughly. 

Author Response

Thanks for the rapid and positive comments.

Round 3

Reviewer 1 Report

Accepted in the present form.

Of note, even if I don't feel qualified to judge about the English language, it seems to me that still some minor spell chek is needed